# Universal vibrational anharmonicity in carbyne-like materials

Johannes M. A. Lechner[1], Pietro Marabotti [1], Lei Shi [2], Thomas Pichler [3], Carlo Spartaco Casari [4] & Sebastian Heeg [1] ✉

Carbyne, an infinite linear chain of carbon atoms, is the truly one-dimensional allotrope of carbon. While ideal carbyne and its fundamental properties have remained elusive, carbyne-like materials such as carbyne chains confined inside carbon nanotubes are available for study. Here, we probe the longitudinal optical phonon (C mode) of confined carbyne chains by Raman spectroscopy up to the third overtone. We observe a strong vibrational anharmonicity that increases with decreasing C mode frequency, reaching up to 8% for the third overtone. Moreover, we find that the relation between vibrational anharmonicity and C mode frequency is universal to carbyne-like materials, including ideal carbyne. This establishes experimentally that carbyne and related materials have pronounced anharmonic potential landscapes which must be included in the theoretical description of their structure and properties.

Carbyne is an infinitely long *sp*-hybridized linear chain of carbon atoms[1,2]. It is the last missing major member in the large family of carbon allotropes. With a cross-section of only one atom, carbyne is the prototypical example of a truly one-dimensional material[1]. As such, it is expected to possess extraordinary properties, with its stiffness, strength, and thermal conductivity exceeding any known material, including other carbon-based materials like carbon nanotubes, graphene, and diamond[2–4]. These prospects have motivated a large amount of theoretical studies on the properties of carbyne and experimental research towards its realization[2,5–12]. Despite the strong and persistent scientific interest, theoretical studies, with, e.g., density functional theory (DFT) calculations, remain challenging, which results in widely varying predictions for the properties of carbyne[6,13–18], e.g., for the band gap, where a range of values between 0.2 and 8 eV has been calculated[14,16]. One of the main reasons for this ambiguity is the difficulty of modeling large systems with conjugated π-bonds by DFT accurately, as pointed out by Johnson et al.[19]. When modeling the potential energy surface of the fundamental stretching vibration of carbyne, DFT studies can even yield both a double and a single well potential depending on the chosen boundary conditions[5]. Experimental data to anchor theoretical calculations and to verify or refute the predicted properties of carbyne is not available to date because, even after decades of efforts, bulk carbyne has not been successfully synthesized.

While the synthesis of bulk carbyne has remained elusive, structurally similar materials exist, among them confined carbyne. It consists of long ( >100 atoms) linear chains of *sp*-hybridized carbon atoms inside multi- or single-walled carbon nanotubes. Following earlier studies on linear carbon chains inside carbon nanotubes[20,21], Shi et al. reported isolated chains of confined carbyne comprising hundreds and thousands of carbon atoms inside double-walled carbon nanotubes (DWCNTs)[22]. As the Peierls theorem predicts, confined carbyne features alternating single and triple bonds. This structure is described by the bond length alternation (BLA), which is defined as the length difference between short and long bonds. Confined carbyne possesses a single Raman active phonon mode, the so-called C mode, a longitudinal optical phonon that corresponds to an oscillation of the BLA along the chain axis. Hence, different BLA values result in different C mode frequencies[23]. This makes Raman spectroscopy the primary experimental method to characterize confined carbyne, aided by the fact that this material possesses the highest Raman cross section ever reported[24]. In general, the C mode appears in the range between 1760

[1]Institut für Physik, Humboldt-Universität zu Berlin, Berlin, Germany. [2]School of Materials Science and Engineering, Sun Yat-sen University, Guangzhou, China. [3]Fakultät für Physik, Universität Wien, Wien, Austria. [4]Dipartimento di Energia, Politecnico di Milano, Milano, Italy. ✉ e-mail: sebastian.heeg@physik.hu-berlin.de

cm⁻¹ and 1870 cm⁻¹ [22,23]. It follows that carbyne chains with different BLAs exist inside carbon nanotubes. The nanotube host prevents the use of alternative optical techniques to characterize confined carbyne, like absorption spectroscopy, due to overlapping resonances, and photoluminescence spectroscopy, due to quenching[25].

A previous extended Raman study on single isolated carbyne chains correlated the properties of a particular carbyne chain with its host nanotube[23]. Confined carbyne chains with a particular C mode frequency (and BLA) always reside inside the same host nanotube chirality. This shows that the BLA and related properties of confined carbyne chains are length-independent but vary among different host nanotube chiralities. In particular, the diameter of the inner host nanotube acts as a parametrization of the properties of the linear carbyne chain[23]. Current strategies for extracting the properties of ideal carbyne from confined carbyne data focus on understanding and eventually factoring out this parametrization by the host nanotube[22,23].

Carbon atomic wires are the second major material class that is structurally similar to carbyne[26,27]. These short linear chains of *sp*-hybridized carbon atoms are terminated by hydrogen or various other end groups. Their properties, such as the frequency of their BLA oscillation, here called ECC mode from effective conjugation coordinate theory, are heavily dependent on the chain length[26]. Hence, for carbon atomic wires, the chain length acts as a parametrization of their properties, similar to the role of the host nanotube in confined carbyne. In addition to Raman spectroscopy, UV-Vis absorption spectroscopy may also be applied in their study[28]. Current strategies for predicting the properties of ideal carbyne from carbon atomic wires focus on understanding the effect of chain length and extrapolating beyond these length dependencies[9,29–31].

While both short carbon atomic wires and confined carbyne possess similar molecular structures, to our knowledge, there has been no attempt to combine their analysis to extract commonalities and unifying principles. This is likely due to the fact that both materials differ in a number of ways, like synthesis methods and stability, and that they are mostly studied in disjunct scientific communities. On top, there is no obvious connecting property beyond a general agreement that both materials are related to ideal carbyne[9,22,28]. One pronounced experimental feature that appears in the Raman spectra of both materials is unusually strong vibrational overtones of the C and ECC modes, respectively[21,28]. Vibrational excitations are usually treated in the harmonic approximation, which yields equidistant quantized energy levels and overtones with frequencies that are exact multiples of the fundamental mode. Real vibrational potentials, however, deviate from the harmonic model. They give rise to anharmonic shifts of the energy levels and overtone frequencies, which often progressively increase for higher-order overtones[32]. Since anharmonicity is directly tied to the shape of the vibrational potential and bond strength, it manifests a fundamental, intrinsic material property. Previous studies have emphasized the importance of anharmonicity in confined carbyne[6,33], but to our knowledge, so far, no attempt has been made to use this property to connect confined carbyne and related materials.

In this work, we establish a universal relationship between geometrical structure and vibrational anharmonicity in one-dimensional *sp*-hybridized carbon, further locking down the properties of ideal carbyne. To this end, we perform Raman spectroscopy of spatially separated confined carbyne chains up to the fourth order of the C mode (i.e., the 4C mode). In contrast to a previous study limited to the second vibrational order[21], we reveal a large degree of vibrational anharmonicity that increases with decreasing C mode frequency. Comparing these results to a similar work by Marabotti et al. on short carbon atomic wires[26], we show that this vibrational anharmonicity is a unifying property of both confined carbyne and short carbon atomic wires by correlating anharmonicity with the material structure. As described previously, confined carbyne and carbon atomic wires are material systems with very different mechanisms of modification - while the BLA strongly varies with the chain length in carbon atomic chains[26], it is solely driven by the interaction with the surrounding nanotube host in confined carbyne and does not depend on the chain length[23]. Therefore, we can exclude a direct influence of these factors and conclude that vibrational anharmonicity is intrinsically controlled by the chain geometry, characterized by the BLA itself. Thus, by combining analysis of confined carbyne and short carbon atomic wires, it is possible to approach and approximate the properties of ideal carbyne.

## Results

We measure the Raman spectra of isolated carbyne chains confined inside DWCNTs as sketched in Fig. 1a. The DWCNTs are dispersed on glass, allowing us to use tip-enhanced Raman spectroscopy to map the chain's C mode, Fig. 1b, and to confirm that the confined carbyne chains are indeed spatially separated. We focus on the Raman spectra of 16 confined carbyne chains with C mode frequencies ($\omega_C$) between 1786–1861 cm⁻¹. This covers around 70% of the entire range of C mode frequencies reported for confined carbyne as a result of parametrization by the host nanotube, and ensures that we pick up any trend that depends on $\omega_C$[4,24,25]. Of these 16 chains, we detect three overtones of the fundamental C mode (i.e., up to the 4C mode) for 11 chains and two overtones (i.e., up to the 3C mode) for the 5 remaining chains. In order to accurately determine the frequencies of the C mode and its overtones, we only consider Raman spectra showing confined carbyne chains with well-resolved, and thus clearly distinguishable peaks and

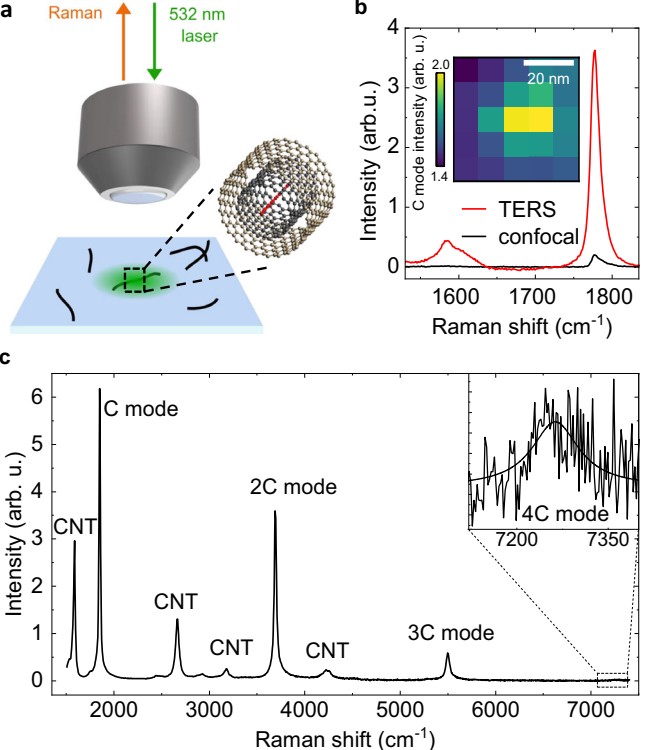

**Fig. 1 | C mode overtone Raman spectrum and TERS characterization of an isolated confined carbyne chain. a** Sketch of confocal Raman measurements on confined carbyne chains inside a double-walled carbon nanotube placed on a glass substrate. **b** Tip-enhanced Raman spectroscopy (TERS) spectrum (red) compared to a confocal Raman spectrum (black) of an isolated confined carbyne chain. Both spectra are taken with 633 nm excitation. Inset: TERS map of C mode, revealing a chain length of ~20 nm. **c** Confocal Raman spectrum of an isolated confined carbyne chain up to the fourth order of the C mode (4C). The 4C mode is magnified 20x due to its low intensity and a Lorentzian fit is included for clarity. The spectrum was taken using a 532 nm laser for excitation. Source data are provided as a Source Data file.

exclude locations where the Raman spectrum shows major contributions of more than two carbyne chains. Exemplary spectra are shown in Suppl. Fig. 1. The corresponding C mode and overtone frequencies are provided in Suppl. Note 4 and selection criteria in Suppl. Note 1 and Suppl. Fig. 2. All Raman spectra are recorded with 532 nm excitation, for which all confined carbyne chains studied here are excited at or close to their fundamental or higher vibronic resonances[18,34].

We show a representative Raman spectrum of an isolated confined carbyne chain in Fig. 1c. The chain's C mode appears as a singular peak at 1861 cm$^{-1}$. It is the only optically active phonon mode of confined carbyne, unlike in short carbon atomic wires, where additional termination-related modes appear[35]. The first, second, and third overtones of the C mode, labeled 2C, 3C, and 4C, appear as approximate frequency multiples of the base C mode frequency. They originate from transitions to the higher energy levels of the chain's vibrational potential. Both the C mode and its overtones are of almost perfect Lorentzian shape such that they can be fitted with an accuracy better than 0.2% of their frequency for all chains and modes reported here. The 2C mode appears at 3702 cm$^{-1}$, the 3C mode at 5505 cm$^{-1}$, and the 4C mode at 7251 cm$^{-1}$. As expected, the Raman intensity decreases with increasing overtone order. While the intensity of the 4$^{th}$ order is rather low compared to the noise level, the peak position can still be determined unambiguously. The clear separation and Lorentzian shape of the identified confined carbyne modes let us conclude that there are no additional spectral features close to these peaks.

We demonstrate the vibrational anharmonicity of confined carbyne by comparing in Fig. 2 the experimental C mode and its overtones, extracted from Fig. 1c, to a hypothetical harmonic Raman spectrum based on a purely harmonic potential (green dashed lines) that scales with the square of a generic vibrational coordinate $q$. In a harmonic potential, vibrational energy levels are equidistant in energy, resulting in equally spaced Raman overtones. Figure 2 displays the increasing redshift between the experimental Raman overtones and the harmonic predictions with mode order, reaching up to 120 cm$^{-1}$ for the 4C mode. This shows the growing impact of the vibrational anharmonicity with higher-order C mode overtones.

The spacing between two consecutive vibrational energy levels is given by $\tilde{\nu}_n - \tilde{\nu}_{n-1}$, where $\tilde{\nu}_n$ is the vibrational frequency of the $n^{th}$ carbyne phonon mode nC. In a harmonic vibrational potential, all spacings between two neighboring energy levels have the same magnitude, yielding the relation $\tilde{\nu}_n - \tilde{\nu}_{n-1} = \tilde{\nu}_1$ for all n. In contrast, in an anharmonic potential with redshifted overtones, we obtain $\tilde{\nu}_n - \tilde{\nu}_{n-1} < \tilde{\nu}_1$. Thus, to quantify the anharmonicity in confined carbyne, we define the anharmonic redshift $\Delta\tilde{\nu}_n$ as $\Delta\tilde{\nu}_n = \tilde{\nu}_1 - (\tilde{\nu}_n - \tilde{\nu}_{n-1})$. We extract $\Delta\tilde{\nu}_n$ from our experimental data and plot it as a function of C mode frequency in Fig. 3a for 11 confined carbyne chains up to the 4C mode and for 5 confined carbyne chains up to the 3C mode. Figure 3a demonstrates the absolute magnitude of vibrational anharmonicity. Ignoring any dependence of $\Delta\tilde{\nu}_n$ on the C mode frequency at this point, we find the expected general increase in anharmonicity with overtone order. However, to compare the anharmonicity of chains with different C mode frequencies, we renormalize the absolute anharmonic shift by the base C mode frequency and define the relative anharmonic redshift $\Delta\tilde{\nu}_{rel,n} = \Delta\tilde{\nu}_n/\tilde{\nu}_1$. $\Delta\tilde{\nu}_{rel,n}$ only depends on the deviation of the shape of the vibrational potential from the harmonic approximation.

To quantify the vibrational anharmonicity in confined carbyne, we plot the relative anharmonic redshift $\Delta\tilde{\nu}_{rel,n}$ as a function of C mode frequency and overtone order for all 16 chains in Fig. 3b and make two key observations. First, the relative anharmonic redshift increases strongly with the mode order, reaching 6.0-7.7% for the 4C mode. These values are up to 8 times larger compared to other solid state systems for which a similar number of overtones is reported (GaN: 0.95%, ZnO: 1.20%, ZnTe: 1.67%)[36–38]. Second, we find a clear trend that

connects the C mode frequency and the anharmonic redshift. Confined carbyne chains with lower C mode frequency possess greater vibrational anharmonicity. The trend is indicated by the dashed lines in Fig. 3b. While the trend is masked for the 2C mode by what appears to be observational random error, in accordance with literature data on the 2C mode of confined carbyne[21,39–43,44], it can be made out in the 3C mode and is clearly pronounced in the 4C anharmonic redshift. The anharmonic redshift increases from around 6.0% for the chain with $\omega_C$ = 1861 cm$^{-1}$ to 7.7% for the chain with $\omega_C$ = 1786 cm$^{-1}$, which represents a relative change of 28%.

Since the C mode frequency of confined carbyne is directly tied to its bond length alternation, and smaller frequencies correspond to a smaller BLA, we conclude that the vibrational anharmonicity in confined carbyne increases with decreasing C mode frequency/BLA. The BLA directly determines a carbyne chain's cumulenic or polyynic character, with a BLA of zero corresponding to a perfect cumulenic state. This means that the more evenly the $\pi$-electrons are distributed along the chain, the higher the vibrational anharmonicity of confined carbyne is.

## Discussion

To gain further insight into the strongly anharmonic behavior observed in confined carbyne's C mode overtones, we model the anharmonic redshift and extract the corresponding anharmonic nondimensional parameter ($\chi$) using second-order vibrational perturbation theory (VPT2) within a single-well potential framework, as formulated by Mendolicchio et al.[45]. VPT2 successfully captures the anharmonic behavior of short carbon atomic wires (i.e., hydrogen-capped polyynes)[26] and, given their structural and vibrational similarities with confined carbyne, we apply the same model to estimate the vibrational anharmonicity of confined carbyne.

We want to particularly emphasize that quantifying anharmonicity with VPT2 within a single-well potential approximation does not infer that carbyne has the same potential shape. Carbyne has been modeled both with single and double minima potential wells by different authors[5,6,15,46]. The corresponding DFT calculations are not conclusive since both models predict unrealistic C mode frequencies far off the experimental data[5,15,46]. Both VPT2 and double-well potential approaches are rooted in expanding the potential up to quartic terms, inherently capturing similar quadratic anharmonic contributions to vibrational energy levels (see Suppl. Notes 2 and 3 in the Supplementary Information)[45,47–50]. Unlike the double-well potential model, which requires complex and computationally demanding calculations, VPT2 provides a direct and analytical method to quantify the anharmonicity from experimental vibrational spectra through the anharmonic parameter $\chi$. Therefore, as our discussion focuses on describing and quantifying the observed anharmonic trend in confined carbyne, we choose a VPT2-based approach while acknowledging that a double-well potential may offer an alternative means to describe carbyne's vibrational properties.

Since confined carbyne features only one Raman-active mode, we simplify the expression of the vibrational energy levels ($E_n$) of ref. 45 and obtain

$$\frac{E_n}{hc} = \varepsilon_0 + \tilde{\nu}_{harm} n - \tilde{\nu}_{harm} \chi (n^2 + n), \tag{1}$$

where $\varepsilon_0$ is the vibrational zero-point frequency (in cm$^{-1}$), $\tilde{\nu}_{harm}$ is the C mode frequency within the harmonic approximation of the potential energy surface (in cm$^{-1}$), and $n$ is the order of the vibrational level. Here, $\tilde{\nu}_{harm}$ represents a renormalization constant to express $\chi$ in nondimensional units. Further details on the derivation of Eqn. (1) are given in Suppl. Note 2 in the Supplementary Information. Before discussing and interpreting our data in detail, we explain the magnitude of the errors in $\chi$ for confined carbyne in Fig. 4, which

appear excessive in relation to the experimental error in Fig. 3. VPT2 provides a quick and direct estimate of vibrational anharmonicity but does not fully account for the stark increase in the anharmonic redshift with overtone order. The resulting systematic fitting error causes the large, uniform uncertainty of $\chi$ in Fig. 4 for confined carbyne. The actual $\chi$ values from VPT2, however, show little variance and reveal a clear correlation of anharmonicity vs. C mode frequency which we discuss in detail further below. Overall, this suggests that our analysis and conclusions are robust to experimental and systematic fitting uncertainties (see Suppl. Note 3 for an extended discussion).

In principle, fourth-order vibrational perturbation theory (VPT4) as established by Gong et al.[51] might yield a better description of anharmonicity in confined carbyne than VPT2, as discussed in detail in

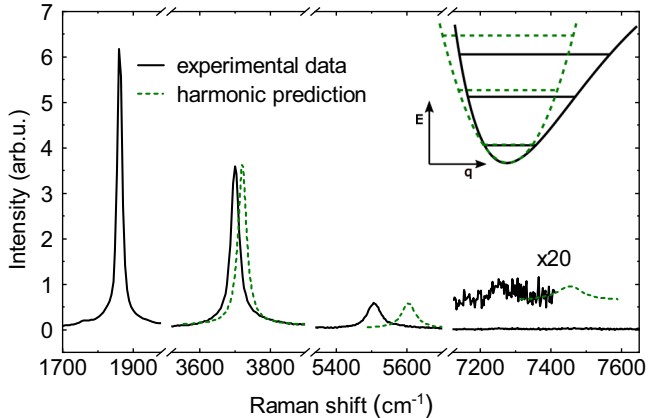

**Fig. 2 | Experimental anharmonic vs. hypothetical harmonic Raman spectrum of C mode and overtones.** Hypothetical spectrum based on a harmonic vibrational potential, compared with an experimental Raman spectrum of a confined carbyne chain with $\omega_C$ = 1861 cm$^{-1}$. The inset shows schematic representations of an anharmonic and a harmonic potential. The potential energy E is given in relation to a generic vibrational coordinate q. Source data are provided as a Source Data file.

Suppl. Note 3 of the Supplementary Information. However, VPT4 uses three fitting variables ($\tilde{\nu}_{harm}$ and two anharmonic parameters), which leads to severe overfitting when applied to the limited number of four data points available per chain in this study, see Suppl. Fig. 4 in the Supplementary Information. Thus, VPT4 is not suitable for modeling our experimental data, but may be an option if Raman measurements that cover additional overtone orders ($n \geq 5$) of confined carbyne become available.

Figure 4 clearly shows that the vibrational anharmonicity of confined carbyne increases as the C mode frequency decreases. This means that the anharmonicity is driven by the BLA, as $\chi$ increases as the BLA (or BLA oscillation frequency) reduces. Previously, we found a roughly linear relationship between the diameter of the innermost host nanotube and the C mode frequency of the confined carbyne chain, where a larger (smaller) inner nanotube diameter corresponds to a higher (lower) C mode frequency, while the confined carbyne chain's length has no effect on its properties[23,25]. Combining this relationship with our results shows that the host nanotubes act as a parametrization of both the chain's BLA and the corresponding vibrational anharmonicity. The host nanotube diameters are provided in Fig. 4 for the relevant C mode range.

Next, we extract the vibrational anharmonicity of short carbon atomic wires from the Raman data reported in ref. 26. The $\chi$ values for carbon atomic wires (red points) are shown in Fig. 4 as a function of their ECC mode[26], which denotes the BLA oscillation in H-capped polyynes. The ECC mode frequency redshifts with increasing chain length, corresponding to a reduction of BLA. Total anharmonic redshift values are shown in Suppl. Fig. 5. As for confined carbyne, the vibrational anharmonicity of carbon atomic wires increases with decreasing BLA/ECC mode frequency, albeit at lower total values than in confined carbyne.

Upon comparing how $\chi$ depends on the C mode for confined carbyne and the ECC mode for carbon atomic wires, we find that the vibrational anharmonicity of the BLA oscillation follows a universal law applicable to both systems. The anharmonicity $\chi$ changes as a linear function of the BLA oscillation frequency $\omega_C$, sketched as a dashed line

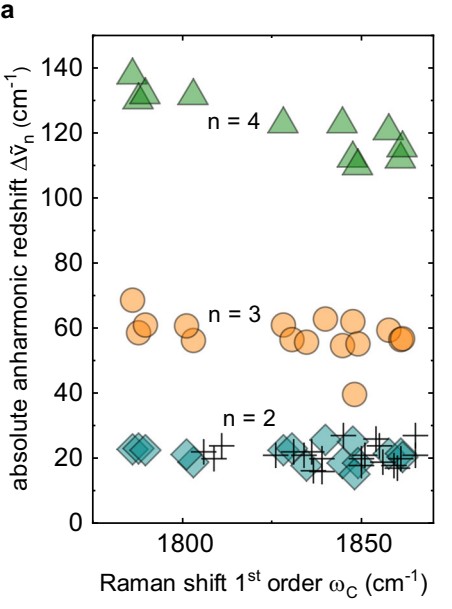
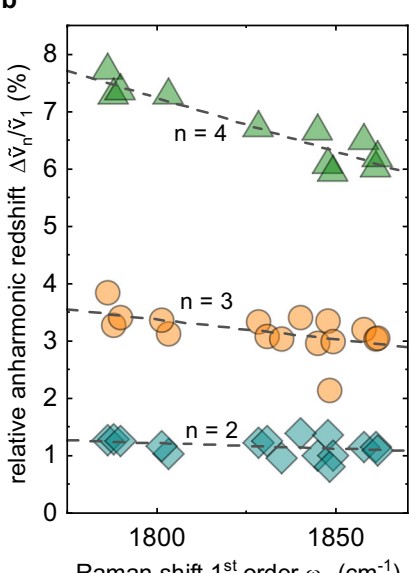
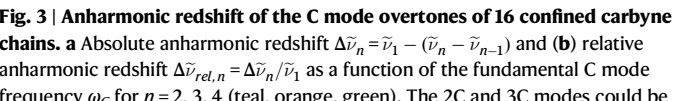

**Fig. 3 | Anharmonic redshift of the C mode overtones of 16 confined carbyne chains. a** Absolute anharmonic redshift $\Delta\tilde{\nu}_n = \tilde{\nu}_1 - (\tilde{\nu}_n - \tilde{\nu}_{n-1})$ and (**b**) relative anharmonic redshift $\Delta\tilde{\nu}_{rel,n} = \Delta\tilde{\nu}_n/\tilde{\nu}_1$ as a function of the fundamental C mode frequency $\omega_C$ for $n$ = 2, 3, 4 (teal, orange, green). The 2C and 3C modes could be

recorded for all chains, the 4C mode for 11 chains. For comparison, redshift values $\Delta\tilde{\nu}_2$ of the 2C mode (crosses) extracted from literature data (refs. 21,39–43,44) are included in (**a**). Source data are provided as a Source Data file.

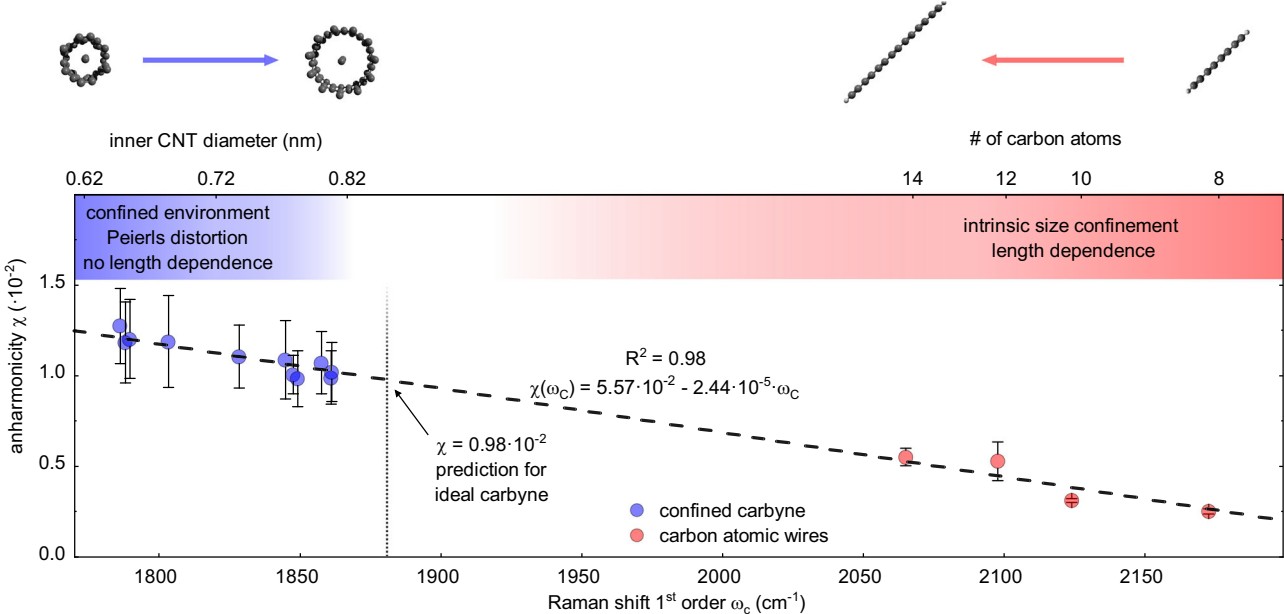

**Fig. 4 | Universal Vibrational Anharmonicity in Carbyne-like Materials.** Anharmonicity ($\chi$) as a linear function of the first-order BLA oscillation Raman mode frequency for confined carbyne (C mode, blue dots) and carbon atomic wires (ECC mode, red dots). The errors derive from the fit of the vibrational energy spacing of confined carbyne and carbon atomic wires using VPT2 (see Eq. S.3 and Suppl. Fig. 3 in Suppl. Note 2). The linear correlation (dashed black line) between $\chi$ and the Raman mode frequency follows Eq. (2). The values of the inner nanotube diameter, which parametrizes the C mode/BLA of confined carbyne chains, are calculated from Eq. 1 in ref. 23. The correlation between the chain lengths of carbon atomic wires and the ECC mode/BLA is described in ref. 26. Source data are provided as a Source Data file.

in Fig. 4, and follows

$$\chi(\omega_C) = b + a \cdot \omega_C. \qquad (2)$$

We obtain $a = -2.44 \pm 0.09[\cdot10^{-5}\, \frac{1}{cm^{-1}}]$, $b = 5.57 \pm 0.18[\cdot10^{-2}]$, and $R^2 = 0.98$, which confirms an excellent fit and the negligible impact of the errors in Fig. 4. This universal dependence of the anharmonicity on the BLA is the first property directly linking confined carbyne chains, governed by the Peierls distortion, and isolated short carbon atomic wires, whose properties are determined by size confinement effects. This shows that a specific environment, such as length confinement or nanotube encapsulation, affects the BLA and BLA oscillation frequency of an *sp*-hybridized structure but does not alter the structure's intrinsic physical properties and their scaling.

Our observations further suggest that any other carbyne-like material systems will show the same dependence of the vibrational anharmonicity on the BLA. The corresponding $\chi$ values can be predicted using Eq. (2) based on the experimentally observed BLA oscillation frequency $\omega_C$. This has several important implications, which we will discuss in the following.

First, we argue that the universal relationship between the vibrational anharmonicity and BLA oscillation frequency is a good measure for experimentally characterizing carbyne-like material systems. If a material shows the vibrational anharmonicity predicted by Eq. (2) in its overtone Raman spectra, it is a strong indication that the material is indeed a carbyne-like material. Beyond anharmonicity, the BLA oscillation frequency $\omega_C$ emerges as a reliable and easily measurable parameter to reveal intrinsic relationships between the properties of a carbyne-like material and its BLA, without the need for direct measurement of the BLA, which requires high-purity crystalline samples. By establishing this relationship, we can infer information about the BLA from easily obtainable Raman measurements, simplifying the study of these materials, irrespective of their specific environment. For instance, if we can relate a specific property of either confined carbyne

chains or short carbon atomic wires to their BLA oscillation frequency, we can, in principle, predict the behavior of all carbyne-like systems.

Second, we can infer the vibrational anharmonicity of ideal carbyne. Since ideal carbyne can be understood both as confined carbyne of infinite diameter host nanotube and as a carbon atomic wire with infinite length, the corresponding $\omega_C$ and $\chi$ values represent upper and lower limits, respectively. A recent experimental work on carbon wires up to 52 atoms long and stabilized by complexation with platinum-based endgroups extrapolated $\omega_C = 1881$ cm$^{-1}$ for ideal carbyne[31], which lies within the expected range. Based on this value, Eq. (2) predicts a vibrational anharmonicity of $\chi = 0.98 \times 10^{-2}$ for ideal carbyne.

Third, we believe that the universal relationship between BLA oscillation frequency and vibrational anharmonicity reported here is of considerable value to theoreticians, providing a basis for refined models and achieving more accurate predictions. The $\chi$ values provided here may serve as a benchmark for theoretical models describing ideal carbyne as well as confined carbyne, carbon atomic wires, and other finite realizations of carbyne. Note that to date, only a few theoretical studies have accounted for anharmonic effects in carbyne[6], while most neglected them, assuming they are small[13,14,16,17]. The comparably large vibrational anharmonicity reported here calls this approach into question.

Fourth, even though the anharmonic parameters in vibrational perturbation theory do not directly correlate to macroscopic material properties, they reveal the limitations of predictions that assumed negligible anharmonicity and provide qualitative trends on how predicted material properties change once anharmonicity is taken into account. Strong vibrational anharmonicity is intrinsically connected with large electron-phonon interactions[52], consistent with the findings of Martinati et al.[34] for confined carbyne and Marabotti et al.[28] for carbon atomic wires. Strong electron-phonon coupling suggests reduced charge carrier mobility in carbyne, which is supported by several experimental and theoretical works[53–58]. Furthermore, strong vibrational anharmonicity is characteristic of strong phonon-phonon scattering, which limits the phonon mean free path and lowers thermal

conductivity[59,60]. This challenges the predicted ultrahigh thermal conductivity of carbyne, which assumes long phonon lifetimes and mean free paths[61]. Strong vibrational anharmonicity implies a reduced bond stiffness, which may be a limiting factor to the exceptional Young's modulus of carbyne calculated within a harmonic framework[3]. This potentially calls the claim that carbyne is the strongest material known into question[62]. Finally, the strong vibrational anharmonicity observed in carbyne may imply higher Grüneisen parameter values than estimated from experimental data using a quasiharmonic model by Costa et al.[63] (0.49 to 0.79). This conclusion is derived from the model applied to flexural modes of graphene by C. H. Lee and C. K. Gan[64]. Future studies will have to consider anharmonic effects and reevaluate how they affect the record-breaking character of the anticipated properties of carbyne.

In conclusion, we have quantified the vibrational anharmonicity of confined carbyne by measuring the Raman modes of single confined carbyne chains up to the third overtone. We find a vibrational anharmonic redshift that increases with overtone order, reaching 8% for the third overtone, and a monotonic increase in vibrational anharmonicity with decreasing C mode frequency. Comparing different confined carbyne chains with short carbon atomic wires of varying lengths, we uncover a universal linear relation between the BLA oscillation frequency and the vibrational anharmonicity of carbyne-like, linear one-dimensional carbon systems. This relationship serves as a fingerprint to identify carbyne-like materials by Raman spectroscopy and acts as an experimental anchor and benchmark for refining theoretical models describing carbyne and related materials. Our work shows that the vibrational anharmonicity of carbyne-like systems cannot be neglected in their theoretical description.

## Methods

### Sample preparation and Raman spectroscopy
Carbyne chains were grown inside double-walled carbon nanotubes in a high-temperature, high-vacuum process as described in ref. 22 and dispersed on a thin glass coverslip. Confocal Raman spectroscopy measurements are conducted with an Xplora Raman spectrometer (Horiba) equipped with a motorized XY-stage using 532 nm excitation. We use acquisition times between 10 s and 150 s, depending on the intensity of the Raman response of specific chains. The applied excitation laser power was kept below 5 mW using an objective with NA = 0.9, corresponding to a power density on the sample of $1.1 \times 10^6$ W/cm$^{-2}$. Following ref. 65, we expect heating of at most 25 K, which corresponds to a heating-induced shift of the C mode smaller than 0.6 cm$^{-1}$. Measuring two chains with very similar C mode frequency (difference < 0.4 cm$^{-1}$) at different powers (1.4 and 4.9 mW respectively), we obtain values of $\chi$ within 3% of each other. This leads us to conclude that heating effects can be neglected. We use a neon glow lamp to calibrate the spectral position data. TERS measurements were conducted with a FabNS Porto-SNOM using a 633 nm He-Ne excitation laser and plasmon-tunable tip pyramids[66].

We determine the Raman shifts of the confined carbyne Raman peaks with Lorentzian fit functions. In order to quantify the anharmonicity in confined carbyne with high precision, only chains with well-resolved peaks with negligible fitting error are included in our analysis. The criteria for this selection are described in detail in Suppl. Note 1 of the Supplementary Information.

### Second-order vibrational perturbation theory to model anharmonicity
In the framework of second-order vibrational perturbation theory (VTP2)[45], anharmonic vibrational levels can be derived by introducing an anharmonic correction matrix ($\chi$). This method describes the perturbed potential energy curve as a fourth-degree polynomial. In the single-mode approximation and considering no mode mixing (as confirmed by our experimental data), valid for the ideal carbyne

system and confined carbyne, the harmonic expression of the potential energy is corrected by adding terms proportional to $q^3$ and $q^4$, where $q$ is the vibrational normal coordinate. The general expression for the vibrational level of quantum number $n$ is expressed as

$$\frac{E_n}{hc} = \varepsilon_0 + \tilde{\nu}_{harm} n - \tilde{\nu}_{harm} \chi (n^2 + n), \tag{3}$$

where $\varepsilon_0$ is the zero-point vibrational energy, $\tilde{\nu}_{harm}$ is the harmonic frequency (in cm$^{-1}$) of the vibrational mode, and $\chi$ is the nondimensional anharmonic parameter. For the difference between the $(n+1)^{th}$ and $n^{th}$ levels, we obtain

$$\frac{E_{n+1} - E_n}{hc} = \tilde{\nu}_{harm}(1 - 2\chi) - (2\chi \tilde{\nu}_{harm})\, n = K - Sn. \tag{4}$$

This relation relies exclusively on parameters that can be determined experimentally by Raman spectroscopy and does not require assuming values that cannot be determined, such as, e.g., the zero-point vibrational energy. Furthermore, we can use Eq. (4) to model our experimental data with a linear regression. From the intercept ($K$) and the slope ($S$), we can calculate $\tilde{\nu}_{harm}$ and $\chi$ as

$$\tilde{\nu}_{harm} = K + S$$
$$\chi = \frac{1}{2(1 + K/S)}. \tag{5}$$

In this work, we compared confined carbyne chains to short carbon atomic wires, whose Raman spectra and the frequencies of their corresponding ECC mode and overtones are displayed in ref. 26.

## Data availability
All data supporting the key findings of this study are shown in the article and the Supplementary Information file and are available at https://doi.org/10.5281/zenodo.15095450. All raw data generated during the current study are available from the corresponding author upon request. Source data are provided with this paper.

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

## Acknowledgements

J.M.A.L., P.M. and S.H. acknowledge funding from the Deutsche Forschungsgemeinschaft (DFG, German Research Foundation) under the Emmy Noether Initiative (Project-ID 433878606). P.M. acknowledges the financial support of the Einstein International Postdoctoral Fellows program (IPF-2022-727). L. S. acknowledges the National Natural Science Foundation of China (52472059). The article processing charge was funded by the Open Access Publication Fund of Humboldt-Universität zu Berlin.

## Author contributions

J.M.A.L. performed the confocal Raman measurements with support from P.M. and evaluated the experimental data. P.M. modeled the vibrational anharmonicity and performed the TERS measurement. L.S. and T.P. provided the confined carbyne sample. C.S.C. assisted in data interpretation and modeling. J.M.A.L., P.M. and S.H. interpreted the data and co-wrote the manuscript with input from all authors. S.H. conceived and supervised the project.

## Funding

## Competing interests

The authors declare no competing interests.
