## [Transparent Peer Review file · Nature Communications]

Universal Vibrational Anharmonicity in Carbyne-like Materials

Corresponding Author: Dr Sebastian Heeg

Version 0:

Reviewer comments:

Reviewer #1

(Remarks to the Author)

The article is to investigate the strong anharmonic behavior observed in confined carbon chain materials, particularly in the C mode of over-vibrating carbon chains. The researchers employed second-order vibrational perturbation theory (VPT2) to model the anharmonic redshift and extract the corresponding anharmonic dimensionless parameter (χ). The study reveals the relationship between vibrational anharmonicity in confined carbon chains and short carbon atomic wires, proposing a universal anharmonicity model applicable to carbon chain materials. This work is interesting and significant; it is recommended for acceptance after addressing the following comments:

Comment1: Although the manuscript discusses the comparison between VPT2 and VPT4, the authors are encouraged to further elaborate on the advantages and disadvantages of these two models.

Comment2: The significant error bars mentioned in Figure 4 require a more detailed explanation. The authors should analyze the sources of these errors and discuss their potential impact on the results.

Comment3: Has the proposed universal relationship been validated in other types of carbon materials? It is recommended that the authors provide additional experimental data or literature support to strengthen the persuasive power of this conclusion.

Comment4: The authors should delve deeper into the physical significance of the results in the discussion section, particularly regarding how vibrational anharmonicity relates to other material properties, such as thermal conductivity and electron mobility.

Comment5: How did the authors select the isolated carbyne chains from DWCNTs? Does the number of the C atoms, or the number of carbyne in DWCNTs influence the experimental results?

Comment6: What is the basis of the hypothetical in Fig.2? Is there any literatures to support this?

Reviewer #2

(Remarks to the Author)

The manuscript "Universal Vibrational Anharmonicity in Carbyne-like Materials" by Lechner and co-authors presents interesting approach and discussion on a highly neglected anharmonicity that is intrinsic to linear atomic chains (be it a carbyne or wire-like structure). The Raman experiments are simple and powerful in the sense that they unambiguously provide means to show how anharmonic a given system is according to its bond length alternation frequencies and overtones. In fact, the use of several overtones to determine anharmonicity was very clever. It is the reviewer's opinion that the manuscript merits publication but before publication some points should be addressed.

01) In the introduction, for example, the authors must mention (or cite) the work by Sharma et al (PRL 125, 105501 (2020)). It is one of the first experimental works to bring up the importance of anharmonicities in the linear chain properties.

02) This work is also important to alert the community that calculations, in special density functional theory, might be many times misleading. This is particularly interesting when considering hydrogen terminated chains (the so-called wires). The authors briefly mention this point in the text (when citing for example refs. 5, 6, 13-18). The experimental results that the authors present seem to refute, for example, the idea of double potential reported in ref. 5. The reviewer suggests that the authors could add a short statement on what is different in their observations and approach in comparison with the DFT results (this could be done either in the main text or in the supplemental information).

03) The reviewer understands that the parameter χ (present in both VPT2 and VPT4 models) and ψ (present in VPT4) are "measures" of anharmonicity but how do they correlate with the anharmonic terms in a potential expansion, for example? Any correlation with possible phonon-phonon interactions? Is it possible to connect these parameters with Young's modulus or Grüneisen parameter? In other words, what is in fact the physical significance of such parameters?

04) The authors claim and show that the VPT2 model does not describe well the experiments, which are properly described by the VPT4 model. Nonetheless the authors decide to use expressions from the model VPT2 with interpretations from the VPT4 model. The reviewer understands the author's attempt with the idea behind this choice but disagrees that the choice is good. In other words, if the VPT4 model is the right model, it should be the one used. This brings the referee back to discussion above in item 3: a change of models changes for example the sign of χ , which is concerning depending on what the answers to item 3 are.

05) Related to the discussion around figure 4: the reviewer understands that the relationship between C-mode frequencies and tube diameters are reported somewhere else but given the importance it apparently has in the results, such relationship should be briefly recalled once again. A point to be clarified: in the text the authors mean to say that the smaller the tube diameter the larger the chain or for a given chain (with a given length) the diameter changes the C-mode frequency? This is an important question: are the authors rendering interactions between tube and chain a second order effect (i.e. negligible)? The model proposed does not consider such interactions, right?

06) Figure S1 (Supplementary material): please, label in the figure what are the nanotube peaks and what are the chain peaks.

07) Equation S4 (Supplementary material): Is it v_{harm} missing before the third and fourth terms?

Reviewer #3

(Remarks to the Author)

This study utilizes Tip-Enhanced Raman Spectroscopy (TERS) to investigate the C-mode and its overtones in 16 distinct CC@DWCNT structures, uncovering the phenomenon of universal vibrational anharmonicity in confined carbyne. This universal anharmonicity can be extended to referenced ECC mode of hydrogen-capped polyynes measured by synchrotron-based UV Raman spectroscopy. The linear relationship between anharmonicity and BLA oscillation Raman mode in 2 different cause of anharmonicity is surprising. The findings demonstrate that vibrational anharmonicity increases with the order of C-mode overtones while decreasing as the C-mode Raman shift increases. The manuscript demonstrates a well-structured and rigorous approach, filling the research gap on the C-mode overtone of confined carbyne. By employing VPT, the study calculates the anharmonicity of various systems and establishes a universal framework for measuring anharmonicity applicable to confined carbyne and carbon atom wires. It provides valuable insights for future research in this area. I think this manuscript can be published in Nature Comm. after considering the following issues.

1. In the discussion section, the manuscript employs the second-order vibrational perturbation theory (VPT2) rather than 4th-order (VPT4). The discussion of complexity of VPT4 is not clear for the reviewer. Clearer comparison of VPT2 and VPT4 is recommended. Several typo of 'VTP2' should be fixed.

2. This study's experimental section discusses the anharmonic redshift of C-mode overtones in different CC@DWCNT systems, with a linear decrease observed as shown in Fig. 3. The decrease is more pronounced for the 4C and 3C overtones, while the 2C data exhibit significant fluctuations around 1850 cm^{-1} , with redshift values greater than those of the data points near 1780 cm^{-1} . How should the authors explain these fluctuations, and is the linear fitting for the 2C state sufficiently rigorous?

3. Regarding the redshift of the 2C mode, previous studies (DOI: 10.1016/j.carbon.2021.07.037, DOI:10.1039/c7nr05883g, DOI:10.48550/arXiv.2411.18899) have reported relevant data for CC@SWCNT systems. The authors could compare these data to evaluate whether they align with the linear relationship proposed in this study.

Version 1:

Reviewer comments:

Reviewer #1

(Remarks to the Author)

The author has well addressed my revision comments. I would suggest acceptance

Reviewer #2

(Remarks to the Author)

The authors have reviewed and answered the reviewer's questions accordingly. It is the reviewer's understanding that the manuscript can now be accepted for publication.

REVIEWER COMMENTS

Reviewer #1 (Remarks to the Author):

The article is to investigate the strong anharmonic behavior observed in confined carbon chain materials, particularly in the C mode of over-vibrating carbon chains. The researchers employed second-order vibrational perturbation theory (VPT2) to model the anharmonic redshift and extract the corresponding anharmonic dimensionless parameter (χ). The study reveals the relationship between vibrational anharmonicity in confined carbon chains and short carbon atomic wires, proposing a universal anharmonicity model applicable to carbon chain materials. This work is interesting and significant; it is recommended for acceptance after addressing the following comments:

We thank the reviewer for the positive evaluation of our manuscript.

Reviewer 1, Question 1: Although the manuscript discusses the comparison between VPT2 and VPT4, the authors are encouraged to further elaborate on the advantages and disadvantages of these two models.

Our Reply: There are a number of considerations to be made when choosing between VPT2 and VPT4. VPT2 is a perturbative correction including the cubic (q^3) and quartic (q^4) power of the vibrational coordinate (q), giving the anharmonic parameter χ . In a simplified picture, the q^3 and q^4 terms reflect three- and four-phonon interactions. VPT4 includes terms up to q^6 , covering more complex interactions, giving the additional anharmonic parameter ψ . Accordingly, when fitting spectral data, VPT2 uses 2 fitting variables (the harmonic vibrational frequency ν_{harm} and χ), while VPT4 uses 3 fitting variables (ν_{harm} , χ , and ψ). This means that, in theory, VPT4 includes nonlinear contributions to the anharmonic redshift, while VPT2 describes a linear increase in anharmonic redshift.

However, a major concern with using VPT4 on our dataset is overfitting. Due to the large energy separation of overtones (~ 230 meV), a maximum of 4 data points per chain can be measured, which are fitted with 3 variables within the VPT4 approximation. A closer inspection of the VPT4 fitting parameters χ , ψ , and ν_{harm} , presented in panels e) to g) of the modified Figure S4 in Supporting Information shown below for convenience, confirms that this concern is justified. For VPT4 there is strong variation in all fitting parameters which does not match the homogenous trend in our experimental data. On the other hand, applying VPT2, ν_{harm} and χ closely follow a linear trend with C mode frequency as would be expected.

Furthermore, the fit residuals (observed value - predicted value) from the VPT4 fits of the anharmonic redshifts in Fig. S4, shown in panel h), show clear signs of overfitting. The residuals are all smaller than the experimental error of around $3\text{-}5\text{ cm}^{-1}$, with some being much smaller (up to three orders of magnitude) to a degree that would be very unlikely in a sufficiently stable model. This renders VPT4 entirely unsuitable to fit our data. For these reasons, we consider VPT2 as the appropriate model to apply to this particular dataset.

Since our additional analysis has led us to the conclusion that since VPT4 is not suitable for our dataset, the result table for the anharmonic parameters fit from VPT4 should be removed

from the supporting information to avoid the impression that they are an accurate description of anharmonicity in confined carbyne.

modified: Figure S.4: VPT4 and VPT2 comparison a)-d) Exemplary fits of individual chains with the VPT2 (upper panels) and VPT4 (lower panels) models. e) χ , f) ψ and g) ν_{harm} fit parameters of confined carbyne chains (blue) and carbon atomic wires (red) as a function of the Raman shift of their corresponding first-order mode according to VPT2 (filled circles) and VPT4 (open circles). The errors derive from the fit (see Eqs. S.5 and S.6). h) Maximum residuals of VPT2 (filled circles) and VPT4 (empty circles) fits as a function of Raman shift of their corresponding first-order mode.

Changes to the manuscript:

- We have modified Figure S.4 in the Supporting Information as shown above to provide evidence that VPT2 is appropriate for describing our data and that VPT4 is not suitable due to overfitting.
- We have removed the tabulated VPT4 fitting parameters from Table S.1 and Fig. S.5 showing VPT4 fits in the Supporting Information.
- As requested by the Reviewer, we now elaborate on the applicability of VPT2 vs. VPT4 in more detail in the manuscript with the changes below.

Changes to manuscript text: “Before discussing and interpreting our data in detail, we explain the magnitude of the errors in χ for confined carbyne in Fig.4, which appear excessive in relation to the experimental error in Fig.4. VPT2 provides a quick and direct estimate of vibrational anharmonicity but does not fully account for the stark increase in the anharmonic redshift with overtone order. The resulting systematic fitting error causes the large, uniform uncertainty in χ Fig.4 for confined carbyne. The actual χ values from VPT2,

however, show little variance and reveal a clear correlation of anharmonicity vs. C mode frequency which we discuss in detail further below. Overall, this suggests that our analysis and conclusions are robust to experimental and systematic fitting uncertainties (see Supporting Information S.3 for an extended discussion).

In principle, fourth-order vibrational perturbation theory (VPT4) as established by Gong et al. might yield a better description of anharmonicity in confined carbyne than VPT2, as discussed in detail in Section S.3 of the Supporting Information. However, VPT4 uses three fitting variables (ν_{harm} and two anharmonic parameters), which leads to severe overfitting when applied to the limited number of four data points available per chain in this study, see Supporting Information. Thus, VPT4 is not suitable for modeling our experimental data but may be an option if Raman measurements that cover additional overtone orders ($n > 4$) of confined carbyne become available.”

Note that the changes above also include changes to the manuscript that pertain to the Question 2 of Reviewer 1

Reviewer 1, Question 2: The significant error bars mentioned in Figure 4 require a more detailed explanation. The authors should analyze the sources of these errors and discuss their potential impact on the results.

Our Reply: The large error bars visible in Figure 4 are a consequence of the applied VPT2 model not being able to fully capture the apparent superlinear increase in anharmonic redshift with mode order, which can be seen in Fig. S.3 of the Supporting Information. As elaborated in our reply to Question 1, using a more complex model like VPT4 results in overfitting due to the limited number of data points available per chain (three fit parameters for four data points). We do not consider the large error bars of the VPT2 analysis to notably impact our conclusions. While for confined carbyne chains the nominal fitting error of the anharmonic parameter χ is larger than expected, the actual calculated values closely follow the observed trend of anharmonicity vs C mode frequency, showing little variance (R^2 value of 0.85 when only considering confined carbyne). This implies that the model is rather robust to experimental error. The negligible impact of the errors for the proposed relation between C mode frequency and anharmonicity expressed by χ is further confirmed when including the comparison with the χ values of the carbon atomic wires. The anharmonic trend line fit over the whole observed C mode range in Fig. 4 possesses an R^2 value of 0.98, which confirms that the model describes the data very well.

Changes to manuscript:

- We include the R^2 value of 0.98 of the linear fit in Fig. 4 to demonstrate the negligible impact of the experimental errors for the proposed relation.
- We now discuss the potential impact of the errors in Fig. 4 in more detail. These aspects are included in the changes to the manuscript in response to the Question 1 of Reviewer 1.

Reviewer 1, Question 3: Has the proposed universal relationship been validated in other types of carbon materials? It is recommended that the authors provide additional experimental data or literature support to strengthen the persuasive power of this conclusion.

Our reply: We thank the reviewer for this comment. Importantly, we do not claim that the relationship we introduce is valid for any other, not carbyne-like carbon materials. The meaning and importance of “universality” in our work are specific to and exclusively valid for carbyne-like materials since it connects two previously separate types of materials, confined carbyne inside carbon nanotubes and carbon atomic wires, through a common “universal” behavior. While confined carbyne has been described as the finite realization of carbyne and as a 1D solid state system, carbon atomic wires were considered molecular systems that shared only a few properties with the “parent” carbyne structure. The consequence of this division is well illustrated in the hypothetical scenario of observing a confined carbyne chain and a carbon atomic wire with the same C mode/ECC mode frequency. Being treated as separate systems, however, this would not imply that they have the same properties such as the degree of (vibrational) anharmonicity, its scaling with vibrational frequency, or other intrinsic properties.

The relationship introduced in this paper links the C mode (or BLA oscillation) frequency to vibrational anharmonicity in a consistent behavior across both carbyne systems. This paves the way for a unified description of confined carbyne and carbon atomic wires as one material class. Naturally, every material shows a consistent, intrinsic relation between bond length and vibrational potential/anharmonicity that is universal for subtypes of the material (i.e., two carbon nanotubes with a different chirality).

To the best of our knowledge, the relation between bond length and vibrational potential/anharmonicity is largely unexplored for carbon nanomaterials such as graphene or carbon nanotubes, for which a division comparable to that of carbyne-like materials has never existed. This means that exploring said relation reveals an intrinsic material property but lacks the unifying character that lies at the core of our work.

Reviewer 1, Question 4: The authors should delve deeper into the physical significance of the results in the discussion section, particularly regarding how vibrational anharmonicity relates to other material properties, such as thermal conductivity and electron mobility.

Our reply: We thank the referee for this remark, which encourages us to lay out the significance of our findings in greater detail. The anharmonic parameters cannot be directly linked in a quantitative way to macroscopic material properties (such as thermal conductivity, electron mobility, Young’s modulus, etc.). They can, however, provide qualitative hints of how properties change compared to previous predictions that assume negligible anharmonicity. Compared to the harmonic approximation, a strong vibrational anharmonicity suggests

1. **a reduced mechanical stiffness and greater sensitivity of vibrational frequencies to volume changes.** A decrease in bond stiffness lowers the Young’s modulus of carbyne compared to the harmonic prediction of 32.71 TPa by Liu et al. (ACS Nano, 7, 11, 10075–10082 (2013), DOI: [10.1021/nn404177r](https://doi.org/10.1021/nn404177r)). This may

challenge the predictions of carbyne being the strongest material known. A phonon or vibration of a higher vibrational anharmonicity suggests higher Grüneisen parameter values. This has been modeled and applied to the flexural modes of graphene in the work of C. H. Lee and C. K. Gan (Phys. Rev. B, 96, 035105 (2017), DOI: [10.1103/PhysRevB.96.035105](https://doi.org/10.1103/PhysRevB.96.035105)).

2. **a reduced charge carrier mobility.** In general, a strong vibrational anharmonicity is intrinsically connected to strong electron-phonon coupling (A. M. Alvertis and E. A. Engel, Phys. Rev. B 105, L180301 (2022), DOI: [10.1103/PhysRevB.105.L180301](https://doi.org/10.1103/PhysRevB.105.L180301)). Strong electron-phonon coupling has already been observed for carbon atomic wires by some of us, Marabotti et al. (ref. 28 in the manuscript), and by Martinati et al. (ref. 34 in the manuscript) for confined carbyne. It further agrees very well with confined carbyne as the strongest Raman scatterer ever reported (ref. 24 in the manuscript). Strong electron-phonon coupling has a negative impact on charge mobility as previously demonstrated for a range of materials in several theoretical and experimental works (e.g., Gunst et al., Phys. Rev. B, 93, 035414 (2016), DOI: [10.1103/PhysRevB.93.035414](https://doi.org/10.1103/PhysRevB.93.035414); A. Wright et al., Nat Commun, 7, 11755 (2016), DOI: [10.1038/ncomms11755](https://doi.org/10.1038/ncomms11755); M. Karakus et al., J. Phys. Chem. Lett., 6, 24, 4991–4996 (2015), DOI: [10.1021/acs.jpcllett.5b02485](https://doi.org/10.1021/acs.jpcllett.5b02485)).
3. **a reduced thermal conductivity.** A strong vibrational anharmonicity is characteristic of strong phonon-phonon scattering. This limits the phonon mean free path and reduces thermal conductivity (see also D. A. Broido et al., Appl. Phys. Lett. 91, 231922 (2007), DOI: [10.1063/1.2822891](https://doi.org/10.1063/1.2822891); G. Pernot et al., Nature Mater 9, 491–495 (2010), DOI: [10.1038/nmat2752](https://doi.org/10.1038/nmat2752)). One of the assumptions leading to the expected ultrahigh thermal conductivity of carbyne are long phonon lifetimes and mean free paths (Ballistic Thermal Transport in Carbyne and Cumulene with Micron-Scale Spectral Acoustic Phonon Mean Free Path. Sci Rep 5, 18122 (2016), DOI: [10.1038/srep18122](https://doi.org/10.1038/srep18122)). This is attributed to intrinsically low phonon-phonon scattering because there is no overlap between the acoustic and optical phonon branches. Our data, however, suggest strong intrinsic phonon-phonon scattering, which will have a negative impact on thermal conductivity.

Despite these qualitative indications, a comprehensive model that quantifies macroscopic parameters directly from vibrational anharmonicity is still lacking. To date, many theoretical studies predicting macroscopic properties of carbyne-like materials assume negligible anharmonicity. We anticipate that our findings will trigger refined theoretical modeling that includes anharmonicity and its expected impact on the (record-breaking) macroscopic properties of carbyne. Experimental data on the electron mobility and thermal conductivity of carbyne or related materials are essentially not available. Once available, the impact of anharmonicity on these parameters can, in principle, be quantified experimentally.

Given these circumstances, we have intentionally kept our discussion of the implications of high vibrational anharmonicity in connection to carbyne's macroscopic parameters more general. Following the suggestion of Reviewer 1, we feel confident to extend our discussion on the potential effects of our findings in the discussion section (pages 15-16):

Changes to the manuscript text: “Fourth, even though the anharmonic parameters in vibrational perturbation theory do not directly correlate to macroscopic material properties, they reveal the limitations of predictions that assumed negligible anharmonicity and provide qualitative trends on how predicated material properties change once anharmonicity is taken into account. Strong vibrational anharmonicity is intrinsically connected with large electron-phonon interactions,⁵¹ consistent with the findings of Martinati et al.³⁴ for confined carbyne and Marabotti et al.²⁸ for carbon atomic wires. Strong electron-phonon coupling suggests reduced charge mobility in carbyne, which is supported by several experimental and theoretical works.⁵²⁻⁵⁷ Furthermore, strong vibrational anharmonicity is characteristic of strong phonon-phonon scattering, which limits the phonon mean free path and lowers thermal conductivity.^{58,59} This challenges the predicted ultrahigh thermal conductivity of carbyne, which assumes long phonon lifetimes and mean free paths.⁶⁰ Strong vibrational anharmonicity implies a reduced bond stiffness, which may be a limiting factor to the exceptional Young’s modulus of carbyne calculated within a harmonic framework.³ This potentially questions the claim that carbyne is the strongest material known.⁶¹ Finally, the strong vibrational anharmonicity observed in carbyne may imply higher Grüneisen parameter values than estimated from experimental data using a quasiharmonic model by Costa et al.⁶² (0.42 to 0.79). This conclusion is derived from the model applied to flexural modes of graphene by C. H. Lee and C. K. Gan.⁶³ Future predictive studies will have to consider anharmonic effects and reevaluate how they affect other the record-breaking character of the anticipated properties of carbyne, such as its record mechanical strength.”

Reviewer 1, Question 5: How did the authors select the isolated carbyne chains from DWCNTs? Does the number of the C atoms, or the number of carbyne in DWCNTs influence the experimental results?

Our Reply: In the literature on Raman spectroscopy of confined carbyne, carbyne chains appear in Raman spectra only with a distinct set of C mode frequencies. A previous study by Heeg et al., Nano Lett., 18, 9, 5426–5431 (2018), <https://doi.org/10.1021/acs.nanolett.8b01681>, revealed that chains inside the same inner host nanotube always have the same C mode frequency. This showed that the chirality of the inner host nanotube determines the C mode of the confined carbyne chain. The set of distinct C mode frequencies for confined carbyne hence represents the number of different distinct nanotube chiralities that are suitable to host carbyne (for an overview, see, i.e., Figure S6 in the Supporting Information of Heeg et al., Nano Lett., 18, 9, 5426–5431 (2018)). It was also shown that there is no dependence of C mode frequency on the length of/number of carbon atoms in the chain. This also means that it is not possible to distinguish between a continuous and a segmented confined carbyne chain inside the same nanotube since both possess the same C mode frequency. This means that the C mode frequency does not change depending on the filling ratio of the host nanotube. Consequently, our results are independent of the number of carbon atoms of a chain/number of chains in a given DWCNT.

The DWCNTs hosting confined carbyne chains are dropcasted on a marked glass slide such that they are spatially separated by the order of micrometers. Since only a small percentage of DWCNTs host carbyne chains, we conduct large hyperspectral spatial maps with a confocal Raman microscope to find confined carbyne chains (see the measurement scheme in Fig. 1a in the manuscript). Confined carbyne chains are identified based on their C mode peak and preselected. Relevant criteria for this selection are the intensity of the C mode

peak, as a high spectral intensity increases the chance of detecting the spectrally weak 3C and 4C modes, which is challenging experimentally, and whether the C mode peak appears to originate from a single chain type.

We then conduct longer spectroscopic measurements on these preselected chains up to the 4C mode. After data acquisition, we fit all accumulated spectra with Lorentzian functions and filter them on the basis of quantitative criteria to limit potential bias. In a first step, we exclude spectra that show major contributions from more than two chains confined in different DWCNTs probed within the laser spot (diameter ~ 1 μm), to decrease fitting uncertainty and prevent erroneous attribution of overtones to the wrong confined carbyne chain. This is considered to be the case if the integrated area of the third-largest Lorentzian fit function used for a C mode or overtone peak is at least 10% as large as the integrated area of the second-largest Lorentzian fit function. Based on this criterion, no spectra with C modes from more than two chains confined in different DWCNTs qualify.

Lastly, we consider the fitting uncertainty corresponding to the spectral position of the Lorentzian fit function of its C mode. If the fitting uncertainty is smaller than 1 cm^{-1} , we include the chain in this study. This value is somewhat arbitrary but comparable to the resolution of scientific-grade Raman spectrometers. Further, this value separates a group of chains with C modes that gave very little fitting uncertainty from chains whose C modes can only be fitted well with larger errors of several cm^{-1} , see Figure S.2 in the Supporting Information.

These criteria yielded 9 locations/spectra with one C mode / single chain type and three locations with two C modes / single chain types.

Reviewer 1, Question 6: What is the basis of the hypothetical in Fig.2? Is there any literatures to support this?

Our reply: We thank the referee for this comment. The hypothetical spectrum displayed in Fig. 2 is based on the harmonic oscillator mode, where vibrational energy levels are equidistant. We used this simplistic model as a baseline to highlight the anharmonic effects observed experimentally. This is a common strategy employed in Raman spectroscopy to promptly quantify the anharmonic shift in overtones and combination modes (see A. Maghsoumi et al., *J. Raman Spec.*, 46, 9 (2015), <https://doi.org/10.1002/jrs.4717>; E. V. Efremov et al., *Anal. Chem.*, 78, 9, 3152–3157 (2006), <https://doi.org/10.1021/ac052253m>; S. V. Krasnoshchekov et al., *Spectrochim. Acta A*, 238, 118396 (2020), <https://doi.org/10.1016/j.saa.2020.118396>; A. Marucci et al., *J. Mater. Res.*, 14, 3447–3454 (1999), <https://doi.org/10.1557/JMR.1999.0466>). Based on the reviewer's comments we extended the description of this aspect in the paper in the paragraph on pages 7-8:

Changes to the Manuscript text: “We demonstrate the vibrational anharmonicity of confined carbyne by comparing in Fig. 2 the **experimental** C mode and its overtones, extracted from Fig. 1c, to a hypothetical ~~entirely~~ Raman spectrum ~~with equidistant peak spacing between two consequent~~ **based on a purely harmonic potential (green dashed lines)**. **In a harmonic potential**, vibrational energy levels are equidistant in energy, resulting in

equally spaced Raman overtones. Fig. 2 displays the increasing separation of the Raman peaks with frequency in the two spectra corresponds to a redshift between the experimentally Raman overtones and the harmonic predictions with mode order, reaching up to 120 cm^{-1} for the 4C mode. This shows the growing impact of the vibrational anharmonicity with higher-order C mode overtones.”

Reviewer #2 (Remarks to the Author):

The manuscript “Universal Vibrational Anharmonicity in Carbyne-like Materials” by Lechner and co-authors presents interesting approach and discussion on a highly neglected anharmonicity that is intrinsic to linear atomic chains (be it a carbyne or wire-like structure). The Raman experiments are simple and powerful in the sense that they unambiguously provide means to show how anharmonic a given system is according to its bond length alternation frequencies and overtones. In fact, the use of several overtones to determine anharmonicity was very clever. It is the reviewer’s opinion that the manuscript merits publication but before publication some points should be addressed.

Reviewer 2, Question 1: In the introduction, for example, the authors must mention (or cite) the work by Sharma et al (PRL 125, 105501 (2020)). It is one of the first experimental works to bring up the importance of anharmonicities in the linear chain properties.

We thank the Reviewer for their recommendation and have included the reference in the introduction (page 4).

Changes to the manuscript text: Since anharmonicity is directly tied to the shape of the vibrational potential and bond strength, it manifests a fundamental, intrinsic material property. Previous studies have emphasized the importance of anharmonicity in confined carbyne, but to our knowledge, so far, no attempt has been made to use this property to connect confined carbyne and related materials.

Reviewer 2, Question 2: This work is also important to alert the community that calculations, in special density functional theory, might be many times misleading. This is particularly interesting when considering hydrogen terminated chains (the so-called wires). The authors briefly mention this point in the text (when citing for example refs. 5, 6, 13-18). The experimental results that the authors present seem to refute, for example, the idea of double potential reported in ref. 5. The reviewer suggests that the authors could add a short statement on what is different in their observations and approach in comparison with the DFT results (this could be done either in the main text or in the supplemental information).

Our reply: We thank the reviewer for this stimulating comment. We are aware that previous works (for example, refs. 5, 6, 15, 45 cited in our manuscript) have suggested the use of a double-well potential to model carbyne vibrational properties, either confined within nanotubes or its ideal representation - i.e., an infinitely long 1D carbon chain. However, most

of these computational results, employing ab initio, Hartree-Fock, or DFT calculations, could not fully capture the experimentally available vibrational data of carbyne. We decided to use vibrational perturbation theory (VPT) within a single-well potential landscape based on the following considerations. First, VPT is a well-established method to quantify anharmonicity directly from experimental vibrational spectra. A VPT-based approach allows us a prompt estimation of vibrational anharmonicity from our unique experimental dataset without requiring any computation effort. In our vision, VPT serves as an initial exploration of anharmonicity in carbyne, providing a baseline for future computational studies.

Second, VPT2 has been successfully used to model the vibrational anharmonicity of carbon atomic wires (ref. 26 in our manuscript). Given the structural and vibrational similarities with confined carbyne, VPT2 could be suitable to capture the anharmonic behavior of confined carbyne.

Third, we note that both VPT2 and an anharmonic double-well potential are rooted in a potential analytic expression that includes up to a quartic power of the vibrational coordinate. With respect to vibrational energy levels, this introduces a similar quadratic anharmonic term which is thus inherently captured by both VPT and an anharmonic double-well potential. Our observations suggest that the vibrational anharmonicity and, in general, the vibrational properties of carbyne can be described by a single-well potential within the framework of VPT. However, our results do not strictly refute the possibility of a double-well potential but rather indicate that a single-well description is sufficient to quantify the observed vibrational anharmonicity of carbyne within the experimental conditions of our study. Further theoretical investigations, based on more accurate and more computationally-demanding models, supported by our experimental dataset, will be necessary to fully explore the impact of a double-well potential on carbyne anharmonicity.

Following the suggestion of the reviewer, we modified the manuscript accordingly to clarify our approach and how it compares to DFT results (pages 10-11).

Changes to the manuscript text: “To gain further insight into the strongly anharmonic behavior observed in confined carbyne’s C mode overtones, we model the anharmonic redshift and extract the corresponding anharmonic nondimensional parameter (χ) using second-order vibrational perturbation theory (VPT2) within a single-well potential framework, as formulated by Mendolicchio et al.⁴⁴ VPT2 successfully captures the anharmonic behavior of short carbon atomic wires (i.e., hydrogen-capped polyynes)²⁶ and, given their structural and vibrational similarity with between confined carbyne and carbon atomic wires, we apply the same model to estimate ~~their~~ vibrational anharmonicity of confined carbyne.”

We want to particularly emphasize that quantifying anharmonicity with VPT2 within a single-well potential approximation does infer that carbyne has the same potential shape. Carbyne has been modeled both with single and double minima potential wells by different authors.^{5,6,15,45} The corresponding DFT calculations are not conclusive since both models predict unrealistic C mode frequencies far off the experimental data.^{5,15,45} Both VPT2 and double-well potential approaches are rooted in expanding the potential up to quartic terms, inherently capturing similar quadratic anharmonic contributions to vibrational energy levels (see Sections S.2 and S.3 in the Supporting Information).^{44,46-49} Unlike the double-well potential model, which requires complex and computationally demanding calculations, VPT2

provides a direct and analytic method to quantify the anharmonicity from experimental vibrational spectra through the anharmonic parameter χ . Therefore, as our discussion focuses on describing and quantifying the observed anharmonic trend in confined carbyne, we choose a VPT2-based approach while acknowledging that a double-well potential may offer an alternative means to describe carbyne's vibrational properties."

Reviewer 2, Question 3: The reviewer understands that the parameter χ (present in both VPT2 and VPT4 models) and ψ (present in VPT4) are "measures" of anharmonicity but how do they correlate with the anharmonic terms in a potential expansion, for example? Any correlation with possible phonon-phonon interactions? Is it possible to connect these parameters with Young's modulus or Grüneisen parameter? In other words, what is in fact the physical significance of such parameters?

Our reply: We thank the reviewer for pointing out the need to expand on the correlation between measures of anharmonicity and material parameters. The anharmonic parameter χ reflects the lower-order (quadratic) anharmonicity contribution to the vibrational energy levels, accounting for q^3 and q^4 terms in the potential expression (where q is the vibrational coordinate). Instead, ψ captures the higher-order (cubic and quartic) nonlinear effects on the vibrational energy levels, corresponding to q^5 and q^6 in the potential expression, within the VPT4 model (J. Z. Gong et al., J. Chem. Phys. 149, 114102 (2018), <https://doi.org/10.1063/1.5040360>). These anharmonic parameters cannot be directly linked in a quantitative way to macroscopic material properties (such as thermal conductivity, electron mobility, Young's modulus, etc.). They can, however, provide qualitative hints of how properties change compared to previous predictions that assume negligible anharmonicity. Compared to the harmonic approximation, a strong vibrational anharmonicity suggests

- 1. a reduced mechanical stiffness and greater sensitivity of vibrational frequencies to volume changes.** A decrease in bond stiffness lowers the Young's modulus of carbyne compared to the harmonic prediction of 32.71 TPa by Liu et al. (ACS Nano, 7, 11, 10075–10082 (2013), DOI: [10.1021/nn404177r](https://doi.org/10.1021/nn404177r)). This may challenge the predictions of carbyne being the strongest material known. A phonon or vibration of a higher vibrational anharmonicity suggests higher Grüneisen parameter values. This has been modeled and applied to the flexural modes of graphene in the work of C. H. Lee and C. K. Gan (Phys. Rev. B, 96, 035105 (2017), DOI: [10.1103/PhysRevB.96.035105](https://doi.org/10.1103/PhysRevB.96.035105)).
- 2. a reduced charge carrier mobility.** In general, a strong vibrational anharmonicity is intrinsically connected to strong electron-phonon coupling (A. M. Alvertis and E. A. Engel, Phys. Rev. B 105, L180301 (2022), DOI: [10.1103/PhysRevB.105.L180301](https://doi.org/10.1103/PhysRevB.105.L180301)). Strong electron-phonon coupling has already been observed for carbon atomic wires by some of us, Marabotti et al. (ref. 28 in the manuscript), and by Martinati et al. (ref. 34 in the manuscript) for confined carbyne. It further agrees very well with confined carbyne as the strongest Raman scatterer ever reported (ref. 24 in the manuscript). Strong electron-phonon coupling has a negative impact on charge mobility as previously demonstrated for a range of materials in several theoretical and experimental works (e.g., Gunst et al., Phys. Rev. B, 93, 035414 (2016), DOI:

[10.1103/PhysRevB.93.035414](https://doi.org/10.1103/PhysRevB.93.035414); A. Wright et al., Nat Commun, 7, 11755 (2016), DOI: [10.1038/ncomms11755](https://doi.org/10.1038/ncomms11755); M. Karakus et al., J. Phys. Chem. Lett., 6, 24, 4991–4996 (2015), DOI: [10.1021/acs.jpcclett.5b02485](https://doi.org/10.1021/acs.jpcclett.5b02485)).

3. **a reduced thermal conductivity.** A strong vibrational anharmonicity is characteristic of strong phonon-phonon scattering. This limits the phonon mean free path and reduces thermal conductivity (see also D. A. Broido et al., Appl. Phys. Lett. 91, 231922 (2007), DOI: [10.1063/1.2822891](https://doi.org/10.1063/1.2822891); G. Pernot et al., Nature Mater 9, 491–495 (2010), DOI: [10.1038/nmat2752](https://doi.org/10.1038/nmat2752)). One of the assumptions leading to the expected ultrahigh thermal conductivity of carbyne are long phonon lifetimes and mean free paths (Ballistic Thermal Transport in Carbyne and Cumulene with Micron-Scale Spectral Acoustic Phonon Mean Free Path. Sci Rep 5, 18122 (2016), DOI: [10.1038/srep18122](https://doi.org/10.1038/srep18122)). This is attributed to intrinsically low phonon-phonon scattering because there is no overlap between the acoustic and optical phonon branches. Our data, however, suggest strong intrinsic phonon-phonon scattering, which will have a negative impact on thermal conductivity.

Despite these qualitative indications, a comprehensive model that quantifies macroscopic parameters directly from vibrational anharmonicity is still lacking. To date, many theoretical studies predicting macroscopic properties of carbyne-like materials assume negligible anharmonicity. We anticipate that our findings will trigger refined theoretical modeling that includes anharmonicity and its expected impact on the (record-breaking) macroscopic properties of carbyne. Experimental data on the electron mobility and thermal conductivity of carbyne or related materials are essentially not available. Once available, the impact of anharmonicity on these parameters can, in principle, be quantified experimentally.

Given these circumstances, we have intentionally kept our discussion of the implications of high vibrational anharmonicity in connection to carbyne's macroscopic parameters more general. Following the supportive suggestion of Reviewer 2, we feel confident to extend our discussion on the potential effects of our findings in the discussion section (pages 15-16):

Changes to the manuscript text: “Fourth, even though the anharmonic parameters in vibrational perturbation theory do not directly correlate to macroscopic material properties, they reveal the limitations of predictions that assumed negligible anharmonicity and provide qualitative trends on how predicated material properties change once anharmonicity is taken into account. Strong vibrational anharmonicity is intrinsically connected with large electron-phonon interactions,⁵¹ consistent with the findings of Martinati et al.³⁴ for confined carbyne and Marabotti et al.²⁸ for carbon atomic wires. Strong electron-phonon coupling suggests reduced charge mobility in carbyne, which is supported by several experimental and theoretical works.⁵²⁻⁵⁷ Furthermore, strong vibrational anharmonicity is characteristic of strong phonon-phonon scattering, which limits the phonon mean free path and lowers thermal conductivity.^{58,59} This challenges the predicted ultrahigh thermal conductivity of carbyne, which assumes long phonon lifetimes and mean free paths.⁶⁰ Strong vibrational anharmonicity implies a reduced bond stiffness, which may be a limiting factor to the exceptional Young's modulus of carbyne calculated within a harmonic framework.³ This potentially questions the claim that carbyne is the strongest material known.⁶¹ Finally, the strong vibrational anharmonicity observed in carbyne may imply higher Grüneisen parameter values than estimated from experimental data using a quasiharmonic model by Costa et al.⁶²

(0.42 to 0.79). This conclusion is derived from the model applied to flexural modes of graphene by C. H. Lee and C. K. Gan.⁶³ Future predictive studies will have to consider anharmonic effects and reevaluate how they affect other the record-breaking character of the anticipated properties of carbyne, such as its record mechanical strength.”

Reviewer 2, Question 4: The authors claim and show that the VPT2 model does not describe well the experiments, which are properly described by the VPT4 model. Nonetheless the authors decide to use expressions from the model VPT2 with interpretations from the VPT4 model. The reviewer understands the author’s attempt with the idea behind this choice but disagrees that the choice is good. In other words, if the VPT4 model is the right model, it should be the one used. This brings the referee back to discussion above in item 3: a change of models changes for example the sign of χ , which is concerning depending on what the answers to item 3 are.

Our Reply: There are a number of considerations to be made when choosing between VPT2 and VPT4. VPT2 is a perturbative correction including the cubic (q^3) and quartic (q^4) power of the vibrational coordinate (q), giving the anharmonic parameter χ . In a simplified picture, the q^3 and q^4 terms reflect three- and four-phonon interactions. VPT4 includes terms up to q^6 , covering more complex interactions, giving the additional anharmonic parameter ψ . Accordingly, when fitting spectral data, VPT2 uses 2 fitting variables (the harmonic vibrational frequency ν_{harm} and χ), while VPT4 uses 3 fitting variables (ν_{harm} , χ , and ψ). This means that, in theory, VPT4 includes nonlinear contributions to the anharmonic redshift, while VPT2 describes a linear increase in anharmonic redshift.

However, a major concern with using VPT4 on our dataset is overfitting. Due to the large energy separation of overtones (~ 230 meV), a maximum of 4 data points per chain can be measured, which are fitted with 3 variables within the VPT4 approximation. A closer inspection of the VPT4 fitting parameters χ , ψ , and ν_{harm} , presented in panels e) to g) of the modified Figure S4 in Supporting Information shown below for convenience, confirms that this concern is justified. For VPT4 there is strong variation in all fitting parameters which does not match the homogenous trend in our experimental data. On the other hand, applying VPT2, ν_{harm} and χ closely follow a linear trend with C mode frequency as would be expected.

Furthermore, the fit residuals (observed value - predicted value) from the VPT4 fits of the anharmonic redshifts in Fig. S4, shown in panel h), show clear signs of overfitting. The residuals are all smaller than the experimental error of around $3\text{-}5\text{ cm}^{-1}$, with some being much smaller (up to three orders of magnitude) to a degree that would be very unlikely in a sufficiently stable model. This renders VPT4 entirely unsuitable to fit our data. For these reasons, we consider VPT2 as the appropriate model to apply to this particular dataset.

Since our additional analysis has led us to the conclusion that since VPT4 is not suitable for our dataset, the result table for the anharmonic parameters fit from VPT4 should be removed from the supporting information to avoid the impression that they are an accurate description of anharmonicity in confined carbyne.

modified: Figure S.4: VPT4 and VPT2 comparison a)-d) Exemplary fits of individual chains with the VPT2 (upper panels) and VPT4 (lower panels) models. e) χ , f) ψ and g) ν_{harm} fit parameters of confined carbyne chains (blue) and carbon atomic wires (red) as a function of the Raman shift of their corresponding first-order mode according to VPT2 (filled circles) and VPT4 (open circles). The errors derive from the fit (see Eqs. S.5 and S.6). h) Maximum residuals of VPT2 (filled circles) and VPT4 (empty circles) fits as a function of Raman shift of their corresponding first-order mode.

Changes to the manuscript:

- We have modified Figure S.4 in the Supporting Information as shown above to provide evidence that VPT2 is appropriate for describing our data and that VPT4 is not suitable due to overfitting.
- we have removed the tabulated VPT4 fitting parameters from Table S.1 and Fig. S.5 showing VPT4 fits in the Supporting Information.
- As requested by the Reviewer, we now elaborate on the applicability of VPT2 vs. VPT4 in more detail in the manuscript with the changes below. Note that this paragraph also includes changes to the manuscript that pertain to the Question 2 of Reviewer 1

Changes to the manuscript text: “Before discussing and interpreting our data in detail, we explain the magnitude of the errors in χ for confined carbyne in Fig.4, which appear excessive in relation to the experimental error in Fig.4. VPT2 provides a quick and direct estimate of vibrational anharmonicity but does not fully account for the stark increase in the anharmonic redshift with overtone order. The resulting systematic fitting error causes the large, uniform uncertainty in χ Fig.4 for confined carbyne. The actual χ values from VPT2, however, show little variance and reveal a clear correlation of anharmonicity vs. C mode

frequency which we discuss in detail further below. Overall, this suggests that our analysis and conclusions are robust to experimental and systematic fitting uncertainties (see Supporting Information S.3 for an extended discussion).

In principle, fourth-order vibrational perturbation theory (VPT4) as established by Gong et al. might yield a better description of anharmonicity in confined carbyne than VPT2, as discussed in detail in Section~S.3 of the Supporting Information. However, VPT4 uses three fitting variables (ν_{harm} and two anharmonic parameters), which leads to severe overfitting when applied to the limited number of four data points available per chain in this study, see Supporting Information. Thus, VPT4 is not suitable for modeling our experimental data but may be an option if Raman measurements that cover additional overtone orders ($n>4$) of confined carbyne become available.”

Reviewer 2, Question 5: Related to the discussion around figure 4: the reviewer understands that the relationship between C-mode frequencies and tube diameters are reported somewhere else but given the importance it apparently has in the results, such relationship should be briefly recalled once again. A point to be clarified: in the text the authors mean to say that the smaller the tube diameter the larger the chain or for a given chain (with a given length) the diameter changes the C-mode frequency? This is an important question: are the authors rendering interactions between tube and chain a second order effect (i.e. negligible)? The model proposed does not consider such interactions, right?

Our Reply: We thank the Reviewer for their recommendation and agree that the relationship between C mode frequencies and host nanotube diameter should be explained in sufficient detail. In essence, we say that the smaller the tube diameter, the smaller the C mode frequency for any confined chain independent of its length.

In the literature on Raman spectroscopy of confined carbyne, carbyne chains appear in Raman spectra only with a distinct set of C mode frequencies. A previous study by Heeg et al., Nano Lett., 18, 9, 5426–5431 (2018), <https://doi.org/10.1021/acs.nanolett.8b01681>, revealed that chains inside the same inner host nanotube always have the same C mode frequency. This showed that it is the chirality of the inner host nanotube that determines the C mode of the confined carbyne chain. The set of distinct C mode frequencies for confined carbyne hence represents the number of different distinct nanotubes chiralities that are suitable to host carbyne (for an overview see, i.e., Figure S6 in the Supporting Information of Heeg et al., Nano Lett., 18, 9, 5426–5431 (2018)).

It was also shown that there is no dependence of C mode frequency on the length of/number of carbon atoms in the chain/number of carbyne chains inside a given nanotube. Therefore, confined carbyne chains of different lengths inside the same nanotube (chirality) have the same C mode frequency. This was confirmed by high-resolution tip-enhanced Raman spectroscopy and supported by the highly unrealistic nature of a scenario where a certain nanotube chirality would exclusively host chains of a specific length. To date, it is not really clear whether the correlation between C mode and nanotube diameter arises from all confined carbyne chains inside nanotubes being long enough to have converged to the limit

of infinite length or if the interaction with the nanotube host largely suppresses any residual length-dependence of the confined carbyne chains.

Importantly, the key aspects of our relation are that the C mode frequency is fundamentally connected to the degree of vibrational anharmonicity and that this correspondence is independent of whether the C mode frequency is determined by intrinsic parameters (length) or by extrinsic interaction (confinement by the nanotube). In this sense, our model is valid without requiring any input or knowledge of the nature of nanotube-chain interaction. In fact, we envision that the universal validity of our model will contribute to a better understanding of nanotube chain interaction and the length dependence for carbon atomic wires without extrinsic confinement.

The question of the reviewer helps to recognize the need to explain this aspect of our work in greater detail which we implement by modifying Fig. 4 and extending our discussion on the relationship between C mode and nanotube diameter.

Changes to the manuscript:

- Figure 4 now contains additional information to clarify the distinction between the length-dependent carbon atomic wires and the length-independent confined carbyne chains controlled by environmental confinement
- The below paragraph on page 13 has been adjusted

Changes to the manuscript text: “Figure 4 clearly shows that the vibrational anharmonicity of confined carbyne increases as the C mode frequency decreases. This means that the anharmonicity is driven by the BLA, as χ increases as the BLA (or BLA oscillation frequency) reduces. Previously, we found a roughly linear relationship between the diameter of the innermost host nanotube and the C mode frequency of the confined carbyne chain, where a larger (smaller) inner nanotube diameter corresponds to a higher (lower) C mode frequency, while the confined carbyne chain's length has no effect on its properties. Combining this relationship with our results shows that the host nanotubes act as a parametrization of both the chain's BLA and the corresponding vibrational anharmonicity. The host nanotube diameters are provided in Fig. 4 for the relevant C mode range.”

Reviewer 2, Question 6: Figure S1 (Supplementary material): please, label in the figure what are the nanotube peaks and what are the chain peaks.

We thank the reviewer for their recommendation and have adjusted the figure accordingly.

Reviewer 2, Question 7: Equation S4 (Supplementary material): Is it v_{harm} missing before the third and fourth terms?

We thank the referee for reporting this oversight to us. We fixed Eq. S.4 by adding v_{harm} before the third and fourth terms.

Reviewer #3 (Remarks to the Author):

This study utilizes Tip-Enhanced Raman Spectroscopy (TERS) to investigate the C-mode and its overtones in 16 distinct CC@DWCNT structures, uncovering the phenomenon of universal vibrational anharmonicity in confined carbyne. This universal anharmonicity can be extended to referenced ECC mode of hydrogen-capped polyynes measured by synchrotron-based UV Raman spectroscopy. The linear relationship between anharmonicity and BLA oscillation Raman mode in 2 different cause of anharmonicity is surprising. The findings demonstrate that vibrational anharmonicity increases with the order of C-mode overtones while decreasing as the C-mode Raman shift increases. The manuscript demonstrates a well-structured and rigorous approach, filling the research gap on the C-mode overtone of confined carbyne. By employing VPT, the study calculates the anharmonicity of various systems and establishes a universal framework for measuring anharmonicity applicable to confined carbyne and carbon atom wires. It provides valuable insights for future research in this area. I think this manuscript can be published in Nature Comm. after considering the following issues.

We thank the Reviewer for recognizing the importance and quality of our work.

Reviewer 3, Question 1: In the discussion section, the manuscript employs the second-order vibrational perturbation theory (VPT2) rather than 4th-order (VPT4). The discussion of complexity of VPT4 is not clear for the reviewer. Clearer comparison of VPT2 and VPT4 is recommended. Several typ of 'VTP2' should be fixed.

Our Reply: There are a number of considerations to be made when choosing between VPT2 and VPT4. VPT2 is a perturbative correction including the cubic (q^3) and quartic (q^4) power of the vibrational coordinate (q), giving the anharmonic parameter χ . In a simplified picture, the q^3 and q^4 terms reflect three- and four-phonon interactions. VPT4 includes terms up to q^6 , covering more complex interactions, giving the additional anharmonic parameter ψ . Accordingly, when fitting spectral data, VPT2 uses 2 fitting variables (the harmonic vibrational frequency ν_{harm} and χ), while VPT4 uses 3 fitting variables (ν_{harm} , χ , and ψ). This means that, in theory, VPT4 includes nonlinear contributions to the anharmonic redshift, while VPT2 describes a linear increase in anharmonic redshift.

However, a major concern with using VPT4 on our dataset is overfitting. Due to the large energy separation of overtones (~ 230 meV), a maximum of 4 data points per chain can be measured, which are fitted with 3 variables within the VPT4 approximation. A closer inspection of the VPT4 fitting parameters χ , ψ , and ν_{harm} , presented in panels e) to g) of the modified Figure S4 in Supporting Information shown below for convenience, confirms that this concern is justified. For VPT4 there is strong variation in all fitting parameters which does not match the homogenous trend in our experimental data. On the other hand, applying VPT2, ν_{harm} and χ closely follow a linear trend with C mode frequency as would be expected.

Furthermore, the fit residuals (observed value - predicted value) from the VPT4 fits of the anharmonic redshifts in Fig. S4, shown in panel h), show clear signs of overfitting. The residuals are all smaller than the experimental error of around $3\text{-}5\text{ cm}^{-1}$, with some being much smaller (up to three orders of magnitude) to a degree that would be very unlikely in a

sufficiently stable model. This renders VPT4 entirely unsuitable to fit our data. For these reasons, we consider VPT2 as the appropriate model to apply to this particular dataset.

Since our additional analysis has led us to the conclusion that since VPT4 is not suitable for our dataset, the result table for the anharmonic parameters fit from VPT4 should be removed from the supporting information to avoid the impression that they are an accurate description of anharmonicity in confined carbyne.

modified: Figure S.4: VPT4 and VPT2 comparison a)-d) Exemplary fits of individual chains with the VPT2 (upper panels) and VPT4 (lower panels) models. e) χ , f) ψ and g) ν_{harm} fit parameters of confined carbyne chains (blue) and carbon atomic wires (red) as a function of the Raman shift of their corresponding first-order mode according to VPT2 (filled circles) and VPT4 (open circles). The errors derive from the fit (see Eqs. S.5 and S.6). h) Maximum residuals of VPT2 (filled circles) and VPT4 (empty circles) fits as a function of Raman shift of their corresponding first-order mode.

Changes to the manuscript:

- We have modified Figure S.4 in the Supporting Information as shown above to provide evidence that VPT2 is appropriate for describing our data and that VPT4 is not suitable due to overfitting.
- we have removed the tabulated VPT4 fitting parameters from Table S.1 and Fig. S.5 showing VPT4 fits in the Supporting Information.
- As requested by the Reviewer, we now elaborate on the applicability of VPT2 vs. VPT4 in more detail in the manuscript with the changes below.

Changes to the manuscript text: “Before discussing and interpreting our data in detail, we explain the magnitude of the errors in χ for confined carbyne in Fig.4, which appear excessive in relation to the experimental error in Fig.4. VPT2 provides a quick and direct estimate of vibrational anharmonicity but does not fully account for the stark increase in the anharmonic redshift with overtone order. The resulting systematic fitting error causes the large, uniform uncertainty in χ Fig.4 for confined carbyne. The actual χ values from VPT2, however, show little variance and reveal a clear correlation of anharmonicity vs. C mode frequency which we discuss in detail further below. Overall, this suggests that our analysis and conclusions are robust to experimental and systematic fitting uncertainties (see Supporting Information S.3 for an extended discussion).

In principle, fourth-order vibrational perturbation theory (VPT4) as established by Gong et al. might yield a better description of anharmonicity in confined carbyne than VPT2, as discussed in detail in Section~S.3 of the Supporting Information. However, VPT4 uses three fitting variables (ν_{harm} and two anharmonic parameters), which leads to severe overfitting when applied to the limited number of four data points available per chain in this study, see Supporting Information. Thus, VPT4 is not suitable for modeling our experimental data but may be an option if Raman measurements that cover additional overtone orders ($n>4$) of confined carbyne become available.”

Note that this paragraph also includes changes to the manuscript that pertain to the Questions 1 & 2 of Reviewer 1

Reviewer 3, Question 2. This study’s experimental section discusses the anharmonic redshift of C-mode overtones in different CC@DWCNT systems, with a linear decrease observed as shown in Fig. 3. The decrease is more pronounced for the 4C and 3C overtones, while the 2C data exhibit significant fluctuations around 1850 cm^{-1} , with redshift values greater than those of the data points near 1780 cm^{-1} . How should the authors explain these fluctuations, and is the linear fitting for the 2C state sufficiently rigorous?

Our reply: The anharmonic redshift is calculated by subtracting the Raman shift of subsequent overtones and comparing it to the Raman shift of the first order or its multiples. This means measuring and fitting uncertainty are compounding, leading to an experimental error of around 3 cm^{-1} , which is notably larger than the expected decrease of the anharmonic redshift of $\sim 2 \text{ cm}^{-1}$ from 1780 cm^{-1} to 1850 cm^{-1} . This amount of variance is common to these types of measurements, as shown by comparison to the corresponding anharmonic redshift extracted from literature data (Refs. 21 and 36 to 41 in the manuscript) of the 1C and 2C modes in confined carbyne presented in the figure below:

2C anharmonic redshift of confined carbyne chains analysed in this study (teal) and of literature data on confined carbyne (crosses) (Refs. 21 and 36 to 41 in the manuscript).

However, the comparison 5 with the anharmonic redshift of the carbon atomic wires as demonstrated in Figure S.6 of the Supporting Information shows that the assumed trendline is justified with an R^2 value of 0.66, increasing to 0.78 when applying weighted linear regression. This value confirms the validity of the model and clearly demonstrates that the linear fitting for the 2C data is in fact sufficiently rigorous.

Reviewer 3, Question 3: Regarding the redshift of the 2C mode, previous studies (DOI: 10.1016/j.carbon.2021.07.037, DOI:10.1039/c7nr05883g, DOI:10.48550/arXiv.2411.18899) have reported relevant data for CC@SWCNT systems. The authors could compare these data to evaluate whether they align with the linear relationship proposed in this study.

Our Reply: We thank the Reviewer for this recommendation. The data that could be extracted from these previous studies with unambiguous respective assignments of 1C and 2C modes indeed aligns well with the linear relationship we have found ourselves (see our reply to Question 2 of Reviewer 3). We have adjusted Figure 3a to include this data.

Changes to the manuscript:

- The main text includes a reference to the literature data.
- Figure 3 (a) now includes the anharmonic redshifts extracted from literature.

Changes to the Manuscript: Figure 3: Anharmonic redshift of the C mode overtones of 16 confined carbyne chains. The 2C and 3C modes could be recorded for all chains, the 4C mode for 11 chains. (a) Absolute anharmonic redshift $\Delta v_n = v_1 - (v_n - v_{n-1})$ and (b) relative anharmonic redshift $\Delta v_{rel,n} = \Delta v_n/v_1$ as a function of the fundamental C mode frequency ω_C for $n=2,3,4$ (turquoise, orange, green). For comparison, redshift values Δv_2 of the 2C mode (crosses) extracted from literature data (Refs. 21, 36–41) are included in (a).